# Left Ventricular Assist Device Pump Obstruction Reduces Native Heart Efficiency

**DOI:** 10.3390/bioengineering10121403

**Published:** 2023-12-07

**Authors:** Ricardo Montes, Saniya Salim Ueckert, Vi Vu, Karen May-Newman

**Affiliations:** Bioengineering Program, San Diego State University, San Diego, CA 92182, USA; ricardo.montes.t@gmail.com (R.M.); salimsd@vcu.edu (S.S.U.); mailtovivu@gmail.com (V.V.)

**Keywords:** heart, ventricle, flow, vortex, LVAD, thrombus

## Abstract

Obstruction of the LVAD flow path can occur when blood clots or tissue overgrowth form within the inflow cannula, pump body, or outflow graft, and it can lead to thrombus, embolism, and stroke. The goal of this study was to measure the impact of progressive pump inflow obstruction on the pressure and flow dynamics of the LVAD-supported heart using a mock circulatory loop. Pump obstruction (PO) was produced by progressively blocking a fraction of the LVAD inlet area. Pressures, flows, and the midplane velocity field of the LV were measured for three LVAD speeds and six PO levels. Pressure and flow decreased with PO, shifting more of the flow through the aortic valve such that the total flow decreased by 6–11% and decreased the efficiency of the work of the native heart up to 60%. PO restricts diastolic flow through the LVAD, which reduces mitral inflow and decreases the strength and energy of the intraventricular vortices. The changes in flow architecture produced by PO include flow stasis and increased shear, which predispose the system to thromboembolic risk. Analysis of the contributions to external work may enable early detection, which allows time for therapeutic intervention, reducing the likelihood of pump replacement and the risk of complications.

## 1. Introduction

Obstruction of the LVAD flow path can occur when blood clots or tissue overgrowth form within the inflow cannula, pump body or outflow graft, and may lead to thrombus, embolism, and stroke [1,2,3,4]. This prepump source of pump obstruction (PO) accounts for 25% of LVAD obstructions, and they form gradually, not suddenly, which makes detection difficult and likely underreported [5,6,7]. Early detection of PO is critical for improving outcomes and relies primarily on changes in LVAD power consumption followed by echocardiographic or computerized tomography evaluation, but they can detect only large obstructions (>75% area) [6,8,9,10]. Innovative strategies such as thromboelastography to assess hypercoagulability and changes in the acoustic vibrations to signal additional inertia within the LVAD have yet to gain widespread traction [11,12]. The mechanical interaction of the native left ventricle (LV) and LVAD is optimized in the hospital by the clinician via the speed settings following a ramp study [13,14]. When the LV beats, pulsatile bypass flow is generated through the LVAD and sometimes through the native aortic valve, redistributing the energy imparted to move blood forward within the circulation. As the LVAD speed increases, the pulsatility of the flow generated by the native heart is reduced by the continuous flow design of the LVAD, decreasing the energy, which plays an important role in maintaining vascular function [15]. PO reduces the cross-section for flow through the LVAD, which results in a higher velocity and shear rate, further exacerbating the risk of thromboembolism [16,17]. The increased resistance through the LVAD affects the LV flow and produces a redistribution and loss of energy in the LV-LVAD system. Our goal in this study was to measure the impact of progressive PO on the pressure and flow dynamics of the LVAD-supported heart using a mock circulatory loop.

## 2. Materials and Methods

A cardiac simulator is a mock loop designed to study the flow conditions of the LVAD-assisted heart. This system has been described in detail in previous publications, a brief summary of which has been provided [18]. A thin, transparent silicone model of a dilated LV was fabricated and assembled with bioprosthetic heart valves in both the aortic and mitral positions. A clear polished version of the HeartMate II LVAD (HM2) inflow cannula was inserted parallel to the septal wall and aimed at the center of the LV with the tip positioned flush with the LV endocardial border [19,20]. The inflow cannula was connected to an HM2, and the LVAD outflow graft was replaced with Tygon tubing connected to the ascending aorta at a 90° angle approximately 10 cm distal to the aortic root. The assembly was immersed in a water-filled tank and attached to a Windkessel model of the circulation [18]. LV pressure, aortic root pressure (AoP), LVAD flow rate (Q_LVAD_), and distal aortic flow rate (Q_Total_) were recorded at 200 Hz (Labchart, AD Instruments, Sydney, Australia). Hydraulic pressure, H, is computed by subtracting LV from AoP, and the flow exiting through the aortic valve is calculated by subtracting Q_LVAD_ from Q_Total_.

A pre-LVAD baseline of a 28% ejection fraction, mean AoP of 65 ± 6 mmHg, and cardiac output of 3.6 ± 0.4 L/min was established to simulate a patient with dilated cardiomyopathy [21]. Thereafter, the resistance and compliance of the circuit remained constant. Throughout the study, the left atrial pressure was maintained at 6 ± 2 mmHg for all conditions. LVAD support was evaluated at three speeds: 8, 9.6, and 11 krpm. PO was modeled with a thin washer placed over the LVAD inlet, as shown in Figure 1. Five different washer sizes were used, each sharing the same inner diameter (2.2 mm) but ranging in outer diameter from 6 to 11 mm. The PO conditions are referred to as the fraction of the original inlet area that was obstructed, with values of 0% (no occlusion), 17%, 39%, 56%, 65%, and 76%. The unobstructed inlet area is 125 mm^2^. All PO levels were studied under the same cardiac waveform and LVAD speed settings of 8, 9.6, and 11 krpm.

The circulating fluid was a viscosity-matched blood analog consisting of 40% glycerol and water (viscosity of 3.72 cP at 20 °C) and was seeded through the left atrial chamber with neutrally buoyant fluorescent tracer particles. Flow visualization was performed with a LaVision Particle Imaging Velocimetry system (LaVision Inc., Ypsilati, MI, USA) using a high-resolution camera (Imager Intense 3) equipped with a wide-angle lens and a fluorescent cutoff filter (450 nm). A laser light sheet 1–2 mm in thickness focused on the model illuminated the particles while the camera captured 12-bit digital images with a time interval of 700–2000 µs. Interrogation windows of 32 × 32 applied to a field of 1376 × 1040 pixels obtained a spatial resolution of 14 pixels/mm. Triggered image pairs were acquired with 50 image sets collected for each time point and phase averaged [18].

### 2.1. Data Analysis

Mean, maximum, and minimum cyclic pressures and flows were averaged over 10 cardiac cycles and used to determine the pulsatility index (PI), as shown in Appendix B. The AV opening time (AVOT) was determined from the time-varying flow signals using methods described previously [23]. The measured variables were analyzed for statistical significance. Normality assessed with the Shapiro-Wilkes test found that the datasets were not normally distributed, and thus, a nonparametric ANOVA (ranking test) was used. The data were analyzed using the open-source software R-code (Lucent Technologies, Murray Hill, NJ, USA) and R Studio. Significance was achieved for *p* ≤ 0.01, and pairwise comparisons were made using the Wilcoxon rank sum test with Bonferroni corrections.

Pulsatile flow through the LVAD generates a counter-clockwise cyclic loop in the hydraulic pressure-flow (H-Q) plot space. A series of equations derived in a previous paper and described in Appendix B were tabulated to identify the contributions to the external work (EW) applied to move blood forward within the cardiovascular system [24].

### 2.2. Image Analysis

A mask corresponding to the model boundary was generated at each time frame and applied to remove the image background. The image field was calibrated using a grid with 2 mm spacing covering the field of view and the 2-D velocity field *u*(*x*,*y*,*t*) obtained for each condition. Regional velocity and pulsatility were calculated for two small (>50 pixels) regions of interest ROI below the mitral valve and at the entrance to the LVAD inflow cannula. Vortex analysis of the measured velocity field data followed our previous methods [18]. The three-dimensional intraventricular vortex ring was visualized in the LV midplane as two vortex cores clockwise (CW) and counter clockwise (CCW). In-plane circulation (Γ), kinetic energy(KE), equivalent radius (R), and aspect ratio (AR) of the tracked vortex cores were determined at each time point during the cardiac cycle.

## 3. Results

The hemodynamics characterized for all conditions are provided in Table 1, with significance noted as asterisks as defined in the legend. At the baseline condition with 0% PO, AoP averaged 86 mmHg for an LVAD speed of 8 krpm, 103 mmHg at 9.6 krpm, and 121 mmHg at 11 krpm. These values were maintained as PO progressed to PO 56%, at which point the values decreased progressively by 10%, 16%, and 21% over the total range for LVAD speeds 8, 9.6, and 11 krpm, respectively. Q_Total_ reached 4.4, 5.0, and 5.5 L/min for the three LVAD speeds with 0% PO and began to fall when PO reached 56%, dropping progressively by 5, 8, and 11%, respectively, for 8, 9.6, and 11 krpm at PO of 76%. Q_LVAD_ reflected a transition from a low bypass flow of 60% at 8 krpm, to 77% at 9.6 krpm, and finally reached 90% at 11 krpm for the 0% PO condition. Over the range of PO, Q_LVAD_ decreased to 39%, 52%, and 62% of the baseline values for 8, 9.6, and 11 krpm, respectively. As the LVAD flow decreased, the flow exiting the LV through the AV increased by 145%, 195%, and 328% at 8 k, 9.6 k, and 11 krpm, respectively. This increase in AV flow was reflected in an increase in AVOT of 13% at 8 krpm, 18% at 9.6 krpm, and 33% at 11 krpm over the range of PO. Greater AV flow was accompanied by an increase in aortic PI of 15% at 8 krpm, 28% at 9.6 krpm, and 46% at 11 krpm. LVAD PI decreased by 25% at 11 krpm and 14% at 9.6 krpm, but there was no difference at 8 krpm. SHE followed the pattern established above, with relatively little change until PO reached 56%, when it began to fall, ultimately decreasing by 46%, 55%, and 56% at 8 k, 9.6 k, and 11 krpm. The phasic differences introduced by cardiac contraction also shifted with PO, as shown in Figure 2. For the unobstructed LVAD inlet, 70% of LVAD flow occurred in systole at 8 krpm, reducing to 60% at higher LVAD speeds. As PO increased, the distribution shifted, with only 55% of LVAD flow occurring in systole at 8 krpm at PO 76%. This fraction decreased further to 45% at 9.6 krpm and 40% at 11 krpm.

When the contraction of the heart is added to continuous flow LVAD support, the H-Q reflects a cyclic loop with the cardiac cycle proceeding along the CCW direction, as shown in Figure 3. The upper left-hand corner corresponds to end diastole, when H is high and Q is low. As the heart contracts, H is reduced and Q increased, with peak Q achieved at end systole. As the pressure relaxes, the heart fills and the cycle begins again. An increase in LVAD speed shifts the curves upward and toward the right, reflecting an increase in the minimum, maximum, and mean flow rate. The corresponding EW_LVAD_ increases with LVAD speed by 18% from 8 to 11 krpm for the unobstructed condition, reflecting the increased contribution of LVAD bypass flow to the total systemic flow. The impact of PO on the H-Q loops is shown in Figure 3 for four selected PO conditions at an LVAD speed of 9.6 krpm. PO conditions below 50% displayed the same H-Q response as the unobstructed condition and are not shown but are reflected in the EW values of Table 2. EW_Total_ increases as more energy is added to the system with increasing LVAD speed but decreases dramatically with increased PO. The efficiency of the heart’s work is diminished as PO progresses, reflected in EW_Heart_LVAD_, which decreases by 60–80% over the full range of PO. The EW associated with flow through the aortic valve includes the work that moves flow forward through the ascending aorta EW_Forward_AA_, which increases with PO, and the energy absorbed from compliance of the aortic root EW_Compliance_AA_, which decreases.

The velocity field measured in the LV midplane in Figure 4 illustrates the vortex ring mitral inflow, diastolic swirling, and systolic outflow as PO increases (See Appendix A). While the main vortex structure of the pattern is maintained, there is a reduction in the strength of the mitral inflow during early diastole (E-wave) and atrial contraction (A-wave), which is confirmed by the results shown in Figure 5. Systolic ejection through the LVAD is reduced with PO, while outflow through the aortic valve increases. As shown in Figure 5, LVAD flow is reduced not only during systole but also in diastole.

Quantitative vortex analysis reflected the vortex ring pattern seen in Figure 4, with a strong CW vortex that moved blood from the mitral valve toward the apex and around toward the aortic valve, while the weaker CCW vortex dissipated along the free wall. Table 3 provides the mean and error values of the vortex parameters, confirming that the circulation and KE of the CW vortex dominate the flow field pattern, with values 5–6 times greater than the CCW vortex. These vortices were elliptical in shape, as determined from the AR, and the diameter of the CW vortex was 2–3 times larger than the CCW vortex. There was a slight trend toward larger and stronger vortices with increasing LVAD speed, but it was small compared to the variance. As PO increased, vortex KE decreased by 20% over the range and circulation by ~10%, but without a notable reduction in the radius.

## 4. Discussion

In this study, pressure and flow decreased with PO at different rates, following a pattern where the resistance to flow through the LVAD increased by >40% at the lowest speed and 25% at the highest, shifting more of the flow through the AV such that the total flow decreased by only 6–11%. However, this redistribution of flow decreased the efficiency of the work of the native heart through the LVAD by up to 60% without a substantial benefit of increased flow through the AV. This study showed that at all LVAD speeds, PO primarily affected diastolic relaxation, restricting LVAD flow, which shifted the H-Q curve toward the left. Systolic contraction was less affected by PO, with a small leftward shift observed only at high PO levels. Improved pulsatility accompanied the increase in AVOT, and there was a substantial overall loss of performance as PO increased. The experimental model of PO progressively reduced the area of the LVAD inlet to <25% of the unobstructed value, and this was accompanied by a 61% reduction in LVAD flow over the range. In addition to low flow, the reduced area resulted in higher velocities through the narrow annular gap remaining open. At all LVAD speeds, the estimated velocities at maximum flow more than doubled over the range of PO, which increased shear on the blood passing through the gap. Combined with the decrease in flow and pulsatility through the LVAD, these flow mechanics greatly increased the risk of thrombus formation within the LVAD.

The indices of external work evaluated in this study reflect changes in the H-Q behavior of the LVAD and the native heart. EW_Heart_LVAD_ is sensitive to PO > 50%, which is similar to other novel approaches, such as accelerometer vibrations [6,25]. The flow mechanics within the LV are altered during LVAD support, with the majority of flow exiting through the apex rather than the native AV. Application of this approach to patients supported with an HM3 necessitates a clear understanding of how to distinguish the impact of the unsynchronized artificial pulse of the LVAD from the changes produced by PO, which have significant overlap in their H-Q behavior [26].

### 4.1. Limitations

A cardiac simulator is designed for experimental consistency as well as flexibility in the use of customized components. The LV geometry captures general trends, not patient-specific conditions, and extrapolation to specific clinical cases warrants caution. The conditions generated by the mock loop encompass a much wider range of values than would be clinically practiced and reflect changes that occur without the effect of autoregulation. The flow patterns are qualitatively similar to patients both pre-LVAD and during LVAD support [18], but are only measured in 2-D under the assumption that the midplane velocity field is representative of an echocardiograph of realistic blood movement in a patient with dilated cardiomyopathy [27]. The disks used to model PO are a rigid material placed over the inlet of the LVAD to obstruct inflow, and they are much stiffer than any thrombus material deposited within the LVAD. However, they also do not impede the rotation of the impeller and thus do not affect the energy required to maintain the LVAD’s speed [22]. The reduction in total systemic flow at the most severe PO level was only 12%, suggesting the model corresponded to a relatively mild degree of pump thrombus.

### 4.2. Conclusions

Despite technical advances, the complications of stroke and pump thrombosis remain serious complications of LVAD implantation [4,28]. Early localized PT detection is crucial for improving outcomes [29,30]. This study demonstrates a new concept for detecting prepump PO, which can detect smaller thrombi than the currently applied index of power consumption. H-Q curves reflect the combined work of the heart and LVAD, which show a marked reduction in area for PO > 50%. Early detection of growing biological masses obstructing the LVAD inflow cannula is critical for reducing LVAD complications but requires a clinical examination and is not routine. Additional software for monitoring the indices of EW would enable onboard continuous assessment to notify patients and their physicians at the earliest sign of PO.

In conclusion, our benchtop model of LVAD pump obstruction demonstrated that PO restricts diastolic flow through the LVAD, which reduces mitral inflow and decreases the strength and energy of the intraventricular vortices. When the native heart contracts, systolic flow through the LVAD is decreased, and some of the energy is used to increase flow and pulsatility through the aortic valve, but much of the external work is lost/dissipated, which is reflected in the H-Q loops. The changes in flow architecture produced by PO include flow stasis and increased shear, which predispose the system to thromboembolic risk [17].

## Figures and Tables

**Figure 1 bioengineering-10-01403-f001:**
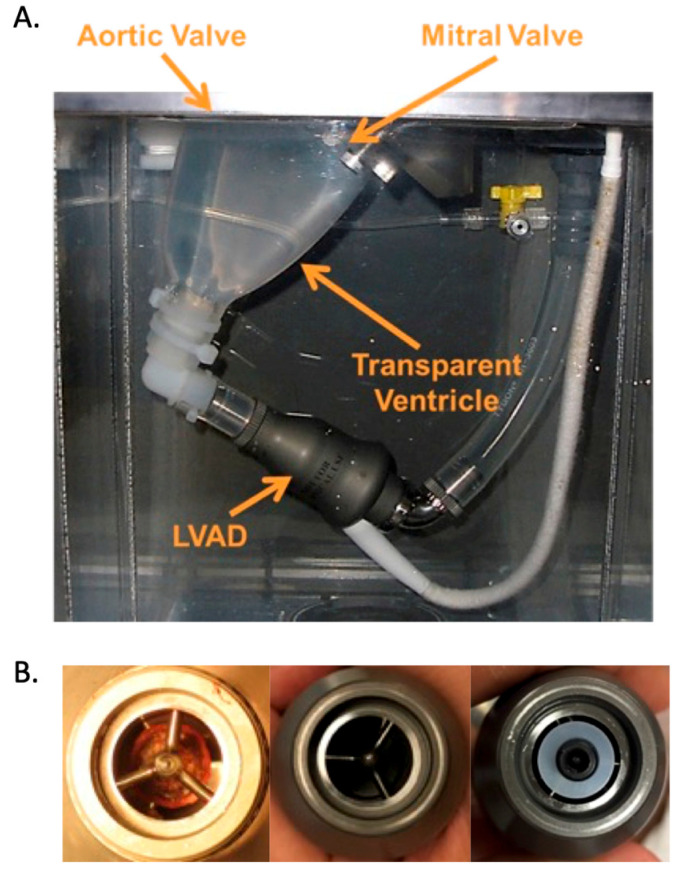
(**A**). Transparent silicone left ventricle model and HeartMate II LVAD configuration inside the mock loop pressure chamber. A transparent inflow cannula protrudes into the ventricle and attaches to the LVAD at the connection where the washer used to create pump obstruction (PO) is placed. (**B**). The left panel shows a pump thrombus removed from an HM2 patient [22]; the middle panel is the unobstructed inlet of the LVAD; the right panel shows a washer positioned over the LVAD inlet to simulate PO.

**Figure 2 bioengineering-10-01403-f002:**
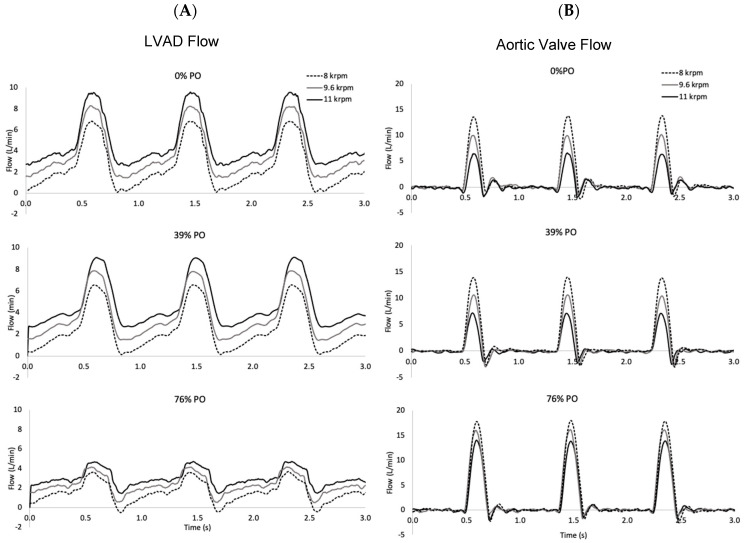
Time series of flow through the LVAD and aortic valve for PO levels of 0%, 39%, and 76% at three LVAD speeds (8, 9.6, and 11 krpm). The native cardiac contraction level corresponds to an unsupported ejection fraction of 21%. (**A**). Higher LVAD speeds result in higher flow rates, but pump obstruction reduces it. (**B**). More flow is ejected through the aortic valve at low LVAD speeds and it increases with the pump obstruction level.

**Figure 3 bioengineering-10-01403-f003:**
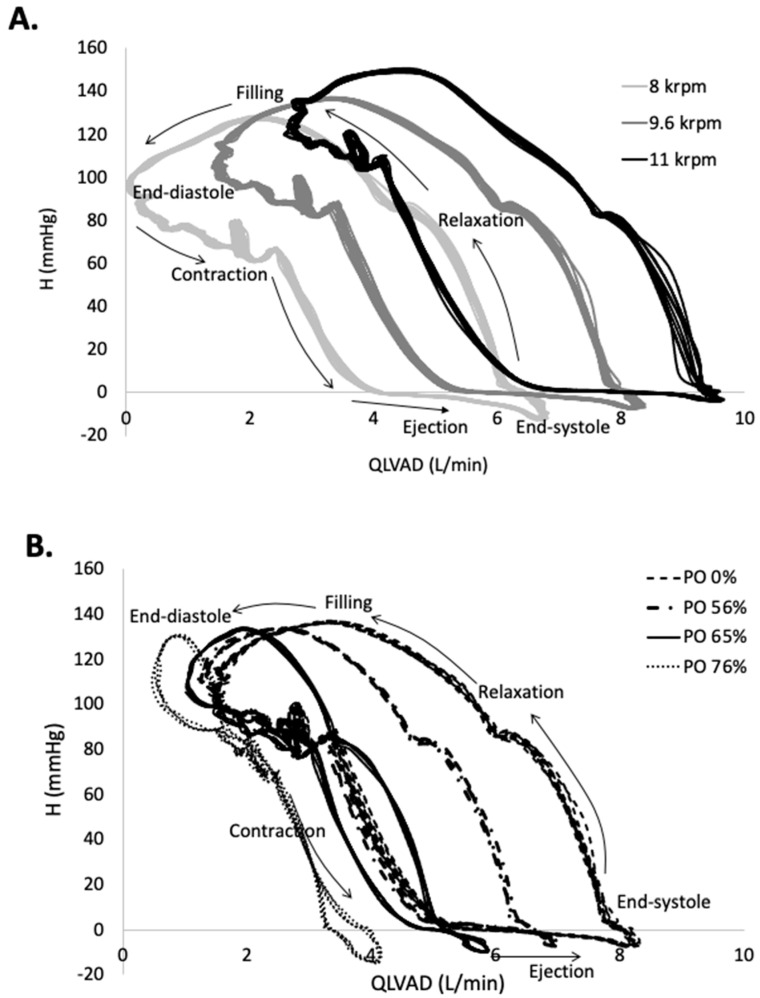
The pressure difference across the LVAD, H, as a function of LVAD flow rate, illustrates the cyclic flow through the LVAD. (**A**). At end diastole, LVAD flow is at a minimum and it increases during the contraction and ejection phases. As LVAD flow increases, the H-Q loop shifts up and to the right. (**B**). As PO is increased from 0% to 76% of the inflow area at a LVAD speed of 9.6 krpm, the H-Q loop shifts leftward and the range of Q decreases dramatically.

**Figure 4 bioengineering-10-01403-f004:**
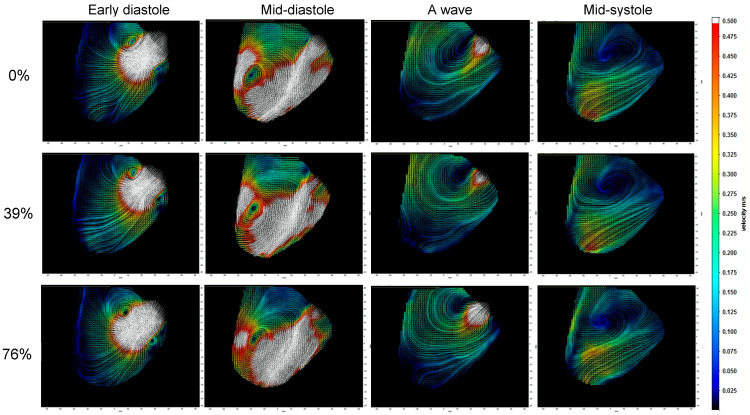
Velocity field images at four points during the cardiac cycle for each condition at a HM2 LVAD speed of 9.6 krpm show how the flow during the cardiac cycle is affected by pump obstruction (PO). In early diastole, the mitral inflow forms a vortex ring visible in the midplane and it grows asymmetrically so that mid-diastole is dominated by the clockwise vortex. The A wave associated with atrial contraction is followed by systolic contraction, which ejects through both the LVAD and the aortic valve. As PO is increased, less flow exits through the LVAD and more through the aortic valve.The color corresponds to velocity magnitude and the field of view is 110 mm (horizontal) by 95 mm (vertical).

**Figure 5 bioengineering-10-01403-f005:**
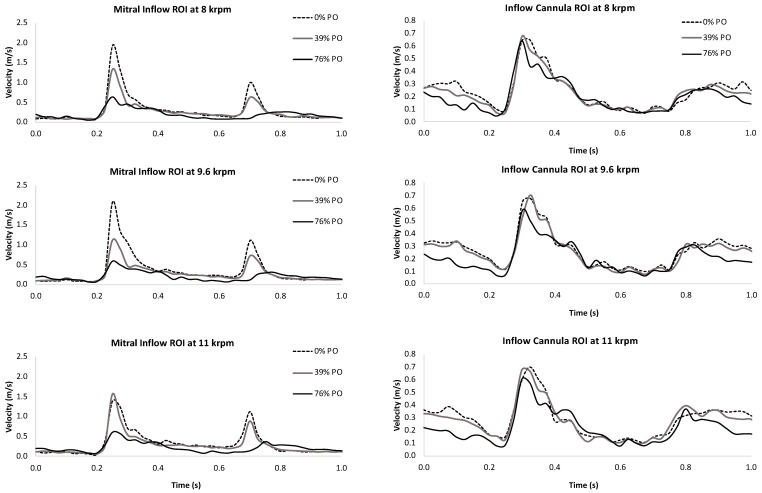
A region of interest positioned over the mitral inlet captured the flow velocity through the mitral valve (**left panels**) and outflow through the LVAD cannula (**right panels**) at 0% (**top**), 39% (**middle**), and 76% (**bottom**) pump obstruction levels for three LVAD speeds. Mitral inflow and LVAD outflow were both dramatically reduced during diastole as pump obstruction increased.

**Table 1 bioengineering-10-01403-t001:** Hemodynamic indices.

PO (%)	LVAD Speed krpm	AoP mmHg	Q_Total_ L/min	Q_LVAD_ L/min	PI Total	PI LVAD	AVOT ms
0%	8	85.9 ± 0.7	4.43 ± 0.16	2.67 ± 0.06	4.56	2.48	188 ± 5
17%	8	86.3 ± 0.6	4.37 ± 0.16	2.60 ± 0.06	4.63	2.50	203 ± 5
39%	8	86.1 ± 0.7	4.40 ± 0.15	2.62 ± 0.05	4.60	2.42	202 ± 5
56%	8	82.8 ± 0.7	4.35 ± 0.16	2.40 ± 0.06	4.89	2.47	198 ± 4
65%	8	81.0 ± 0.7	4.26 ± 0.16	2.17 ± 0.04	5.16	2.42	205 ± 7
76%	8	77.3 ± 0.7	4.18 ± 0.17	1.63 ± 0.03	5.26	2.55	212 ± 7
0%	9.6	103.2 ± 0.7	5.01 ± 0.13	3.88 ± 0.06	3.34	1.75	175 ± 5
17%	9.6	103.6 ± 0.6	4.97 ± 0.13	3.79 ± 0.06	3.38	1.75	187 ± 5
39%	9.6	103.0 ± 0.6	4.96 ± 0.13	3.80 ± 0.06	3.40	1.66	188 ± 5
56%	9.6	98.7 ± 0.6	4.89 ± 0.14	3.51 ± 0.05	3.63	1.62	190 ± 5
65%	9.6	95.1 ± 0.6	4.76 ± 0.14	3.14 ± 0.04	3.94	1.53	197 ± 8
76%	9.6	87.2 ± 0.7	4.59 ± 0.16	2.38 ± 0.03	4.27	1.50	207 ± 8
0%	11	121.0 ± 0.5	5.55 ± 0.10	4.98 ± 0.06	2.41	1.40	152 ± 5
17%	11	121.2 ± 0.5	5.51 ± 0.10	4.85 ± 0.06	2.45	1.39	167 ± 6
39%	11	120.3 ± 0.4	5.52 ± 0.10	4.87 ± 0.06	2.46	1.31	167 ± 5
56%	11	114.4 ± 0.5	5.37 ± 0.12	4.50 ± 0.05	2.72	1.23	170 ± 6
65%	11	108.4 ± 0.5	5.20 ± 0.12	3.99 ± 0.04	3.05	1.12	185 ± 7
76%	11	95.5 ± 0.6	4.92 ± 0.14	3.03 ± 0.02	3.51	1.06	202 ± 8

**Table 2 bioengineering-10-01403-t002:** External work (in J) calculated from the experimental data.

PO (%)	LVAD Speed (krpm)	EW_Total_ (J)	EW_Heart_ (J)	EW_LVAD_ (J)	EW_Heart_LVAD_ (J)	EW_Forward_AA_ (J)	EW_Compliance_AA_ (J)
0%	8	1.116	1.016	0.316	0.891	0.125	−0.216
17%	8	0.926	1.097	0.316	1.005	0.091	−0.487
39%	8	0.943	1.065	0.316	0.926	0.138	−0.438
56%	8	0.301	0.510	0.316	0.358	0.152	−0.525
65%	8	0.097	0.234	0.316	0.142	0.092	−0.453
76%	8	0.173	0.232	0.316	0.111	0.121	−0.375
0%	9.6	1.455	1.001	0.582	0.937	0.064	−0.128
17%	9.6	1.353	1.192	0.582	1.144	0.048	−0.421
39%	9.6	1.266	1.068	0.582	1.068	0.022	−0.384
56%	9.6	0.718	0.666	0.582	0.569	0.097	−0.530
65%	9.6	0.500	0.411	0.582	0.311	0.100	−0.492
76%	9.6	0.573	0.167	0.582	0.047	0.119	−0.175
0%	11	1.792	1.020	0.845	0.997	0.023	−0.073
17%	11	1.764	1.287	0.845	1.286	0.001	−0.368
39%	11	1.710	1.147	0.845	1.144	0.003	−0.282
56%	11	1.027	0.636	0.845	0.562	0.073	−0.454
65%	11	0.749	0.354	0.845	0.274	0.081	−0.451
76%	11	0.723	0.081	0.845	0.005	0.092	−0.203

**Table 3 bioengineering-10-01403-t003:** LV vortex characteristics.

PO (%)	LVAD Speed krpm	Γ_CW_ ×10^−3^ m^2^/s	Γ_CCW_ ×10^−3^ m^2^/s	KE_CW_ J/m	KE_CCW_ J/m	R_CW_ cm	R_CCW_ cm	AR_CW_	AR_CCW_
0%	8	27.4 ± 17.5	−5.5 ± 8.3	41.9 ± 57.4	10.4 ± 18.6	0.87 ± 0.18	0.34 ± 0.10	1.85 ± 0.43	1.61 ± 0.54
17%	8	29.1 ± 16.4	−5.3 ± 7.6	37.4 ± 37.3	8.9 ± 16.4	0.88 ± 0.20	0.34 ± 0.10	1.70 ± 0.44	1.52 ± 0.90
39%	8	28.6 ± 15.8	−6.0 ± 8.0	38.4 ± 44.2	11.2 ± 17.8	0.88 ± 0.18	0.42 ± 0.20	1.83 ± 0.51	1.65 ± 0.60
56%	8	28.1 ± 15.9	−5.3 ± 6.9	36.0 ± 38.6	10.1 ± 17.3	0.87 ± 0.17	0.38 ± 0.12	1.72 ± 0.39	1.56 ± 0.47
65%	8	27.3 ± 15.3	−5.2 ± 7.2	35.2 ± 37.9	9.4 ± 15.5	0.91 ± 0.21	0.35 ± 0.15	1.74 ± 0.39	1.69 ± 0.88
76%	8	25.5 ± 16.4	−4.6 ± 6.9	33.8 ± 43.0	8.1 ± 15.8	0.88 ± 0.20	0.35 ± 0.10	1.78 ± 0.44	1.45 ± 0.90
0%	9.6	29.8 ± 17.4	−5.7 ± 10.2	43.5 ± 56.3	14.5 ± 35.0	0.89 ± 0.20	0.39 ± 0.14	1.81 ± 0.43	1.66 ± 0.68
17%	9.6	28.3 ± 18.6	−6.4 ± 8.6	37.3 ± 39.8	12.4 ± 18.9	0.89 ± 0.16	0.45 ± 0.39	1.76 ± 0.47	1.68 ± 0.40
39%	9.6	29.7 ± 16.3	−5.9 ± 8.0	39.2 ± 41.4	12.6 ± 20.7	0.88 ± 0.20	0.4 ± 0.13	1.68 ± 0.37	1.58 ± 0.53
56%	9.6	29.1 ± 16.7	−5.5 ± 7.6	37.6 ± 40.7	9.8 ± 15.4	0.89 ± 0.17	0.43 ± 0.14	1.8 0± 0.47	1.89 ± 0.94
65%	9.6	27.9 ± 15.4	−5.5 ± 7.5	36.9 ± 45.9	9.5 ± 18.2	0.89 ± 0.17	0.37 ± 0.17	1.73 ± 0.39	1.68 ± 0.59
76%	9.6	26.3 ± 16.3	−5.1 ± 7.1	35.6 ± 45.4	9.7 ± 15.3	0.87 ± 0.16	0.43 ± 0.39	1.83 ± 0.47	1.68 ± 0.40
0%	11	30.2 ± 18.2	−5.2 ± 8.9	41.7 ± 50.6	11.6 ± 22.9	0.93 ± 0.18	0.44 ± 0.17	1.82 ± 0.43	1.93 ± 0.68
17%	11	29.5 ± 17.8	−6.3 ± 8.2	38.2 ± 40.4	11.4 ± 17.5	0.91 ± 0.20	0.43 ± 0.22	1.73 ± 0.46	1.59 ± 0.45
39%	11	28.8 ± 17.7	−5.4 ± 8.6	42.1 ± 50.7	11.3 ± 19.6	0.89 ± 0.17	0.37 ± 0.17	1.74 ± 0.47	1.62 ± 0.57
56%	11	28.5 ± 16.6	−5.3 ± 8.9	37.7 ± 42.0	9.9 ± 23.6	0.88 ± 0.18	0.41 ± 0.19	1.74 ± 0.52	1.91 ± 0.68
65%	11	27.8 ± 15.9	−5.0 ± 6.7	36.2 ± 43.4	9.3 ± 14.7	0.92 ± 0.21	0.33 ± 0.13	1.77 ± 0.49	1.77 ± 0.70
76%	11	26.5 ± 17.1	−5.3 ± 7.6	37.1 ± 46.5	10.1 ± 16.2	0.89 ± 0.20	0.42 ± 0.22	1.89 ± 0.46	1.82 ± 0.45

Mean ± standard deviation over the cardiac cycle for vortex circulation (Γ), kinetic energy (KE), equivalent radius (R), and aspect ratio (AR) for clockwise (CW) and counter-clockwise (CCW) vortices.

## Data Availability

The sensor and image data are available upon request.

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
