# Peer review of "Left Ventricular Assist Device Pump Obstruction Reduces Native Heart Efficiency"

_bioengineering, 2023, doi:10.3390/bioengineering10121403_

Round 1

Reviewer 1 Report

Comments and Suggestions for Authors

Very well and accurately documented and designed experiment, what I lack is the clinical relevance of the results.

Comments on the Quality of English Language

Please reformulate the sentence at lines 215,216

At line 260, I would suggest : "..........is used to increase........"

Author Response

Reviewer 1

Very well and accurately documented and designed experiment, what I lack is the clinical relevance of the results.

This point has been addressed by revisions of the introduction and discussion that provide more background on the current clinical approach and its shortcomings. An alternate accelrometer technology currently in development was also cited for earlier detection of pump obstruction, as it has the same sensitivity as the H-Q method described in this manuscript.

At line 260, I would suggest : "..........is used to increase........"

This sentence has been revised.

Reviewer 2 Report

Comments and Suggestions for Authors

This is a study aimed at simulating in a mock circulatory loop the changes induced by progressive degrees of pump obstruction (PO) of axial flow LVAD systems such as the HM2 on the pressure and flow dynamics of the LVAD-supported heart, with the aim of obtaining innovative indices capable of detecting PO early with greater sensitivity and specificity than the echocardiographic ones currently used.

Although they are still a long way from obtaining usable results in a clinical model, the authors present interesting data and discuss potentially promising implications for future studies.

It would be interesting if, after examining the limitations of the study, the authors focused on future prospects, briefly indicating what changes they intend to make to their circulatory model, in order to take into account some variables that they themselves identify and which characterize the clinical context of an LVAD supported heart of a patient with dilated heart disease.

Author Response

Reviewer 2

This is a study aimed at simulating in a mock circulatory loop the changes induced by progressive degrees of pump obstruction (PO) of axial flow LVAD systems such as the HM2 on the pressure and flow dynamics of the LVAD-supported heart, with the aim of obtaining innovative indices capable of detecting PO early with greater sensitivity and specificity than the echocardiographic ones currently used.

Although they are still a long way from obtaining usable results in a clinical model, the authors present interesting data and discuss potentially promising implications for future studies.

It would be interesting if, after examining the limitations of the study, the authors focused on future prospects, briefly indicating what changes they intend to make to their circulatory model, in order to take into account some variables that they themselves identify and which characterize the clinical context of an LVAD supported heart of a patient with dilated heart disease.

The approach to detecting PO using external work could be implemented using software that monitors more time-varying features of the LVAD and potentially be continuous and connected to the provider. An alternative accelerometer approach is also cited that demonstrates similar sensitivity to PO. The introduction and discussion were both revised to provide more clinical context and relevance.

Reviewer 3 Report

Comments and Suggestions for Authors

The authors provide an in-vitro model to study the effects of thrombus formation and tissue growth on a left ventricular assist device (LVAD) and how thrombus formation and tissue growth change the flow patterns in the left ventricle model. 

The manuscript needs some minor revisions:

Several acronyms are used without specifying them; it would be good to provide a table that states the acronyms and their complete forms. 

It would be good to provide illustrations of the LVDA model with different closures for readers to understand easily.

All image panels need further explanations (in the images themselves), and the graphs and Images need x-labels in the same direction.

It would be great to show the statistics of number of patients with LVAD who develop thrombus and tissue outgrowth.

Author Response

Reviewer 3

The authors provide an in-vitro model to study the effects of thrombus formation and tissue growth on a left ventricular assist device (LVAD) and how thrombus formation and tissue growth change the flow patterns in the left ventricle model. 

The manuscript needs some minor revisions:

Several acronyms are used without specifying them; it would be good to provide a table that states the acronyms and their complete forms. 

It would be good to provide illustrations of the LVDA model with different closures for readers to understand easily.

All image panels need further explanations (in the images themselves), and the graphs and Images need x-labels in the same direction.

It would be great to show the statistics of number of patients with LVAD who develop thrombus and tissue outgrowth.

  • A list of acronyms has been provided.
  • The figure showing the pump obstruction has been revised, as well as the figure legend (Figure 1).
  • This has been completed.
  • The level of detail on the number of patients with inflow cannula obstruction is not available, but estimate indicate that 25% of pump malfunction due to obstruction are due to this source. While the number of LVAD patients remains small compared to other cardiovascular implants, the severity of complications is very high - stroke or death. Thus, there is a need to lower the risk of all contributing factors.